# Bioactivity Potential of Bioceramic-Based Root Canal Sealers: A Scoping Review

**DOI:** 10.3390/life12111853

**Published:** 2022-11-11

**Authors:** Mauro Schmitz Estivalet, Lucas Peixoto de Araújo, Felipe Immich, Adriana Fernandes da Silva, Nadia de Souza Ferreira, Wellington Luiz de Oliveira da Rosa, Evandro Piva

**Affiliations:** 1Program in Dentistry, School of Dentistry, Federal University of Pelotas (UFPEL), Pelotas 96010-610, RS, Brazil; 2Department of Restorative Dentistry, Division of Endodontics, Piracicaba Dental School, State University of Campinas (UNICAMP), Piracicaba 13083-970, SP, Brazil; 3Department of Restorative Dentistry, School of Dentistry, Federal University of Pelotas (UFPEL), Pelotas 96015-560, RS, Brazil; 4Department of Semiology and Clinics, School of Dentistry, Federal University of Pelotas (UFPEL), Pelotas 96015-560, RS, Brazil

**Keywords:** calcium-silicate, bioactivity, bioceramics, root canal sealer, endodontics, scoping review

## Abstract

Introduction: Bioceramic-based root canal sealers are novel materials with a bioactivity potential that stands out compared with conventional root canal sealers. However, the term bioactivity may be overused and is often misunderstood. Hence, the objective of this study was to synthesize and map key concepts related to the bioactivity analysis of bioceramic-based root canal sealers. Methods: The present scoping review is reported in accordance with the PRISMA-ScR Statement and is registered in the Open Science Framework. Two blinded reviewers carried out a comprehensive search in six databases up to January 10th, 2022: MEDLINE, Scopus, Embase, Web of Science, Cochrane Library, and Lilacs/BBO. Eligibility was considered for in vitro and in vivo studies that evaluated the bioactivity potential of bioceramic-based root canal sealers. Results: A total of 53 studies were included in the qualitative synthesis. In vitro bioactivity was evaluated through the mineralization potential, formation of carbonated apatite on the surface, and the gene expression related to proteins involved in the mineralization process. Meanwhile, for in vivo studies, staining techniques associated with immunohistochemical tests were mainly used to detect mineralization on the material–host tissue interface. Conclusions: According to the methodology used, the most prevalent methods to assess bioactivity in acellular form were the immersion of the material in Hank’s balanced salt solution, followed by surface observation with scanning electron microscopy and energy dispersive X-ray. In cell cultures, the chosen method was usually Alizarin Red staining, followed by the evaluation of alkaline phosphatase enzymatic activity and the use of molecular biology tests.

## 1. Introduction

The filling of the root canal system is an important step in endodontic treatment and has been attributed to the fact that it fosters the sealing of both the main root canal and its accessory ramifications, thus preventing the transit of microorganisms between the root canal and the periradicular tissues [1]. The most commonly used materials for filling root canals are gutta-percha associated with a root canal sealer [2]. Ideally, the root canal sealer should adhere to the root canal walls, promote adequate sealing, have an adequate radiopacity, have low setting contraction, have thin and small particles, not stain the dental tissues, have antimicrobial activity, be biocompatible, be insoluble to tissue fluids, have a workable setting time, and be easily removable from the inside of the root canal system if necessary [3,4].

Root canal sealers can be classified according to their composition as zinc-oxide-eugenol-based, calcium-hydroxide-based, resin-based, glass-ionomer-based, silicon-based, and bioceramic-based [5]. In 2007, a calcium-silicate-based root canal sealer was developed and launched on the market under the commercial name iRoot SP^®^ (Innovate Bioceramix, Vancouver, BC, Canada) and was classified as a bioceramic-based root canal sealer [6].

Bioceramics are ceramic materials that contain silica, alumina, zirconia, bioactive glasses, ceramic glasses, calcium silicates, hydroxyapatite, and calcium phosphate [7]. For the filling of the root canal system, the goal of using hydraulic types of sealer is to achieve the most hermetic filling possible with inert materials. Additionally, the bioactivity potential and adhesion to the dentinal substrate of these types of materials are desirable characteristics [8].

Calcium-silicate-based root canal sealers show promising results when evaluated in vitro, surpassing conventional root canal sealers regarding some of their properties [9]. Bioceramic-based root canal sealers stand out for their biological characteristics when applied in various clinical situations, especially in cases where the risk of material extrusion into the periodontium is greater, such as in cases of root resorption, teeth with open apex, or overinstrumented canals [10].

When discussing biological properties in dental materials, some terminology should be considered. Biocompatibility is defined as “the ability of a material to function with an appropriate host response in a specific application” [11]. Meanwhile, the term bioactivity has been widely used in the market as a characteristic of bioceramic materials, showing a lack of standardization in the literature regarding the concept definition and the methodologies needed to assess this characteristic in dental materials. Bioactivity can be defined as the cellular response induced by the release of ions or biologically active substances from the biomaterial so that biomineralization occurs [12]. From this perspective, it was suggested to limit the term bioactive to materials that encourage biomineralization specifically, rather than the material in vitro performance [13]. However, this definition differs in a few studies [14,15].

For these reasons, the objective of this scoping review was to synthesize and map key concepts from studies that evaluated the bioactivity potential of bioceramic root canal sealers for better use of this concept in future studies. Thus, the following exploratory research question was asked based on the population–concept–context (PCC) framework for scoping reviews: What are the main methodologies (C) used to determine the bioactivity potential (C) of bioceramic-based root canal sealers (P)?

## 2. Methodology

### 2.1. Protocol Registration

The protocol for this scoping review is available on the Open Science Framework through the link https://osf.io/3jdqu/ (acessed on 17 May 2021). The reporting of this study followed the recommendations of the Preferred Reporting Items for Systematic Review and Meta-analysis extension for Scoping Reviews (PRISMA-ScR) [16,17].

### 2.2. Eligibility and Exclusion Criteria

The inclusion and exclusion criteria were established by consensus among the reviewers after an in-depth discussion considering the research question, the study objectives, and possible methodological limitations.

Eligible studies were those that evaluated the bioactivity potential of bioceramic-based root canal sealers (commercially available or experimental) used in association with gutta-percha. In vitro studies that evaluated the bioactivity potential of these root canal sealers were included. Furthermore, in vivo biocompatibility studies were also included. No restriction on the publication date of the evaluated studies was applied, but language restrictions were applied for English and Spanish.

Studies that evaluated only cytotoxicity, cell viability, or cell proliferation were excluded from this scoping review, as well as biocompatibility studies evaluating only the inflammatory response. In addition, studies that evaluated only root perforation or apical surgery were excluded.

### 2.3. Search Strategy

A search strategy was formulated using Medical Subject Headings terms (MeSH), Emtree terms, and Health Science Descriptors (DeCS) related to the research question, and is available in Appendix A. The terms used were selected to cover the largest number of relevant studies. An initial systematic search was performed by two independent and blinded reviewers (M.S.E. and L.P.A.) on 8 May 2021, in the following electronic databases: Cochrane Library, EMBASE, LILACS/BBO, PubMed/MEDLINE, SciVerse Scopus, and the Web of Science. This initial search aimed to verify whether the selected terms were in accordance with the study’s objective and eligibility criteria.

### 2.4. Selection Process

After a systematic search in the databases, the studies’ references were imported into Mendeley software (Elsevier, Amsterdam, The Netherlands) for duplicate removal. Then, they were exported into Rayyan online software (Qatar Computing Research Institute, Doha, Qatar) [18], where screening by title and abstract was performed by two independent and blinded reviewers (M.S.E. and L.P.A.). After the selection process, blinding between the reviewers was removed, and in the case of any disagreement, a third researcher with more experience in the field (E.P.) settled the issue. The selection was made according to the eligibility criteria and the articles that met the inclusion criteria or those with insufficient data in the title and abstract to make a clear decision were selected for full-text analysis. Additionally, references from the included studies were manually searched to identify potential studies that were not covered by the search strategy. 

### 2.5. Data Collection Process

Data collection was performed by two independent and blinded reviewers (M.S.E. and L.P.A.) in an Excel spreadsheet (Microsoft Corporation, Redmond, WA, USA) with data regarding the author’s name, year of publication, published journal, root canal sealer used, dilution medium (if applied), control group, bioactivity analysis method, sample number, duration (time of evaluation), and main findings. After the initial data collection, a third reviewer (F.I.) double-checked all of the retrieved data to avoid tabulation errors. This model was made in accordance with the recommendation for scoping reviews by the Joanna Briggs Institute [19] and was adjusted and previously tested by all of the reviewers involved in this study with at least two included articles. No quality assessment of the included evidence was carried out, as the aim of this review was to map the investigation techniques used to evaluate the bioactivity potential of bioceramic-based root canal sealers.

## 3. Results

The systematic search (last conducted on 10 January 2022) retrieved 3883 potentially relevant studies. Figure 1 is a schematic flowchart that synthesizes the article selection process according to the PRISMA 2020 Statement [17]. After removing duplicate records, 2218 studies were screened by title and abstract using the Rayyan online application (Qatar Computing Research Institute). A total of 2143 studies were excluded for not meeting the inclusion criteria and 75 studies were selected for full-text analysis. Of the 75 studies, 22 were excluded for the following reasons: 2 evaluated the bioactivity potential of restorative bioceramic materials; 8 studies evaluated the bioactivity of root canal sealers that do not have calcium silicate in their formulation; 1 was a narrative literature review not related to the bioactivity of root canal sealers; and, in 11 studies, the bioactivity potential was not evaluated. The remaining 53 studies met all inclusion criteria and were included in this scoping review.

### Study Characteristics

A total of 53 studies evaluated the bioactivity potential of bioceramic-based root canal sealers, of which 37 were in vitro studies, and the remaining 16 were in vivo studies. The studies investigated 21 commercial and 17 experimental formulations among the different bioceramic-based root canal sealers. Although MTA Fillapex is based on salicylate resin, it was included in this analysis because it has 13% of MTA (mineral trioxide aggregate) in its composition, has a different biological and physicochemical behavior from conventional root canal sealers, and was classified as a bioceramic in a previous study [20].

The included studies had a divergence regarding the evaluation of the acellular and cellular bioactivity potential owing to the different methodologies used to investigate this property. This review classified acellular and cellular in vitro and in vivo studies separately. Among the included in vitro studies, 12 of them performed acellular in vitro investigations by immersing the hardened material in solutions that simulate body fluids (e.g., Hank’s balanced salt solution, phosphate buffered saline, and simulated body fluid) followed by the material surface observation, with two of these studies evaluating the dentin-filling material interface. Among the solutions used in the methodologies, Hank’s balanced salt solution (HBSS) was the most frequently used, with seven studies, followed by simulated body fluid (SBF) with three studies and phosphate buffered saline (PBS) solution with two studies. After immersing the specimens in these solutions, different methods were used to observe the surface morphology deposited on the materials’ surface or the interface with the dentin. A swept emission field electron microscope (FE-SEM) was used in three studies, while eight used scanning electron microscopy (SEM) and one study used scanning electron microscopy with electron probe microanalysis (SEM-EPMA) for joint evaluation of microscopy and chemical or structural analysis. To evaluate the elemental composition of the material deposited on the surface of the bioceramic-based root canal sealers, it was also used in seven studies. Energy-dispersive X-ray spectroscopy (EDS or EDX) and X-ray diffraction (XRD) were used in three studies and Fourier transform infrared spectroscopy (FT-IR) was used in two studies. SEM-EPMA (scanning electron microscopy-electron probe microanalysis), micro-Raman spectroscopy, and confocal laser microscopy were used in one study each. Table 1 highlights the bioactivity analysis method used for each study.

All of the studies included in this review have observed material deposition along the filling surface or mineralization reactions at the interface with the root dentin, and these findings suggest the bioactivity potential of bioceramic-based root canal sealers.

Of the included studies, 28 evaluated the bioactivity potential in cells (Table 2), with 21 in human cells and 7 studies in animal origin cells. Of the studies with human cells origin, 20 were cell lines originating from the apical region (cells of the periodontal ligament, tooth germ, apical papilla, and osteoblasts). The methods most used to assess the bioactivity potential in cells were the enzymatic activity of Alizarin Phosphatase (*n* = 20), followed by the use of Alizarin Red staining (*n* = 16), molecular biology tests (e.g., RT–PCR and RT–qPCR) (*n* = 16), enzyme-linked immunosorbent assay (ELISA) (*n* = 5), the Von Kossa technique (*n* = 3), immunofluorescence techniques (*n* = 2), immunocytochemical and confocal laser microscopy, cell adhesion, and scanning electron microscopy (*n* = 1 each).

Alkaline phosphatase enzyme (ALP) was evaluated in 19 studies through an enzymatic assay [21,32,34,38,44,48,50,54], RT–PCR [37,41], RT–qPCR [25,39,40,45,46,47,52], colorimetric method [23,40,52], and ELISA [35,52]. In 12 of these studies, positive results were observed regarding the bioactivity potential for all of the tested bioceramic-based root canal sealers. A recent study [56] evaluated the immune bioactivity of BioRoot-RCS and found the evidences of the potential of upregulation and immunomodulatory properties for cytokine production involved in healing process and regeneration of periapical lesions. Moreover, formulations containing hydroxyapatite supplemented with calcium silicate cementent have been tested [57]. However, in other studies, materials such MTA Fillapex [44], TECHBiosealer [39], experimental calcium silicate sealer [32,50], Neo MTA Plus, experimental tricalcium silicate sealer, and TotalFill BC Sealer [54] showed no biological effect.

The Alizarin red staining method was used by 16 studies to evaluate the mineralization nodules’ neoformation [23,25,34,37,40,41,42,45,47,49,50,52,53,54,56]. Real-time polymerase chain reaction (RT–qPCR) was used in nine studies [25,38,39,40,45,46,47,52,56] for the detection of proteins linked to mineralization, while reverse transcription polymerase chain reaction (RT–PCR) was used in another five studies [23,34,37,41,53] for the detection of proteins linked to bone repair or angiogenesis.

The most evaluated biological markers evaluated in the included studies were as follows: ALP in 11 studies; runt-related transcription factor 2 (RUNX2) in 7 studies; osteocalcin (OCN) in 6 studies; CEMP-1 and CAP in 4 studies each; DMP-1 and OPN in 3 studies each; Osterix-2 and DSPP-2 in 2 studies each; and ON, β-actin, Phex-1, AMBN, and AMELX in 1 study each.

The Endosequence BC Sealer was the bioceramic-based root canal sealer that obtained the most favorable results in five studies, along with iRoot^®^ SP and MTA Fillapex^®^ in three studies each; TotalFill BC and BioRoot RCS in two studies each; Ceraseal, Endoseal and Endosequence Hi-Flow in two studies each; and Bio-C sealer, Bio-C Sealer ION+, Well-Root ST, C-Root, Neo MTA Plus^®^, and the experimental sealers showed positive results in one study each.

It is also relevant to acknowledge that one study [35] was carried out with the ex vivo use of rat’s parietal bone to evaluate the bone tissue response to the tested bioceramic-based root canal sealers.

Of the 16 included in vivo studies shown in Table 3, nine evaluated only the reaction of rats’ subcutaneous tissue o the materials, while four studies evaluated the bone implantation in rats, one concerned the evaluation of bone implantation in a rabbit, one study investigated both bone and subcutaneous tissue implantation in rats, and one investigated root canal treatment performed in dogs with radiographic and histological evaluation. One study evaluated bone implantation and the periapical tissue response by implanting bioceramic material in periapical tissues and not on the femur or tibia [58]. In 14 studies, the tested materials had a positive result for bioactive potential and, in 1 study, partial formation of bone tissue was observed in the period evaluated [59]. Additionally, nine studies used the von Kossa histochemical technique to detect mineralization activity [60,61,62,63,64,65,66,67].

In general, the analysis of the apatite deposition in vitro of materials can be carried out by evaluating the mechanism of apatite formation with immersion in phosphate-rich solutions and observation of the deposited material and further evaluating the materials reactivity, with this method being reported in some studies as a partial assessment of the bioactivity potential or acellular bioactivity [23].

The mineralized nodules’ detection in cell cultures is mainly observed in a direct way by the staining technique using Alizarin red dye and indirectly by the enzymatic activity of the enzyme alkaline phosphatase and by the detection of genes or substances involved in biomineralization. The combination of these assessments at the cellular level also results in a partial assessment of the bioactivity potential, called cellular bioactivity [23].

For in vivo studies evaluating bioceramic-based root canal sealers, concerning the materials’ subcutaneous implantation, the most mentioned detection methods for evaluating mineralized tissue formation were the use of histochemical hematoxylin–eosin techniques associated with the use of the Von Kossa staining technique [60,61,64,66,69], immunohistochemistry techniques to detect osteocalcin activity [65,66], osteopontin calcein detection with the use of Alizarin red dye [67], and usually a combination of those methods. When the in vivo evaluation was in bone tissue, histochemical analysis with the use of hematoxylin–eosin was preferred among the studies.

## 4. Discussion

Dental materials’ bioactivity is a desired property for root canal sealers and is continually sought after by researchers and the dental materials industry to obtain a higher tissue repair rate, especially with the formation of mineralized tissue [70]. The term bioactivity has a broad definition, with the term bioactive referring to a material that was designed to induce a specific biological response [74]. However, depending on the area, a biomaterial can also be one that exhibits tissue adhesion as a result of the interaction with tissues at the interface [75]. Based on ISO 23317:2014 [76], for biomaterials that are implanted in a living body, a thin layer of calcium and phosphorous will form on its surface, then this apatite layer connects the implanted biomaterial to the living tissue without a distinct boundary.

Although the focus of this study is on the bioactivity of bioceramic-based root canal sealers, it is not just in the area of endodontics that bioactivity is a desired property. With the growth in the search for conservative and minimally invasive dentistry, several dentistry areas have sought bioactivity in their materials, such as bone implants [77], bone grafts [78], pulp capping agents [79], and restorative materials [80]. More recently, researchers have come to the concept that a bioactive material is one that acts upon or interacts with living cells and tissues to produce a specific response, such as biologically directed mineral formation [81]. This is a very desirable property in root canal sealers as dentin is vulnerable to hydrolyses, which can be catalyzed by stages of endodontic treatment, such as the use of sodium hypochlorite, calcium hydroxide dressings, and the fluids’ penetration and diffusion [82].

There was significant variation in the parameters used to indicate bioactivity in the included studies. Several methodologies were found to assess this property in bioceramic-based root canal sealers, and the goal of most of them was to induce the carbonated apatite formation on the materials surface, mineralization nodules’ formation in cells, and in vivo mineralized tissue formation.

Regarding the methodologies used to assess the potential for acellular bioactivity, immersion in phosphate-rich solutions refers to the material ability to induce in vitro apatite formation on the surface. This methodology is of interest because it is similar to in vivo tests and reduces the need for unnecessary animal testing [83]. Other solutions that simulate body fluids have been suggested for the same purpose, such as phosphate-buffered saline and Hank’s balanced saline solution [26]. For the assessment of bioactivity in bioactive glasses, this methodology was referred to as an initial evaluation for the development of new materials and a partial bioactivity analysis because it assesses the material reactivity. The material reactivity is related to the release of calcium ions from the material, which react with the solution phosphate ions to form a calcium phosphate precipitate (similar to hydroxyapatite) on the material surface, not involving a biological reaction [27,84].

When evaluating the bioactivity potential in cells, the most frequent evaluation was using the enzyme alkaline phosphatase (ALP), followed by the technique using Alizarin Red (ARS) and molecular biology (RT–PCR and RT–qPCR). This is similar to another literature review that evaluated the bioceramic materials’ bioactivity for pulp capping in human pulp stem cells (hDPSCs), in which the most frequent method was RT–PCR, followed by ALP and ARS [85]. In another study, it was reported that the bioactive potential of calcium-silicate-based materials could be assessed by osteogenic differentiation and mineralization by measuring ALP activity using ARS and the gene expression of genes linked to mineralization [86].

Furthermore, in studies with cells, the parameters to assess bioactivity were cell adhesion [15] and gene expression of proteins related to bone repair [37]. The ALP enzymatic activity was evaluated using dyes for evidence of mineralization, which is one of the main substances used to evaluate the bioactivity in cells according to the frequency presented in the included studies. Of the 20 studies evaluating this enzyme, 16 showed positive results, 2 were nonsignificant, and 2 had no positive effects.

In recent studies, the designation of material with bioactivity potential (acellular and cellular) suggests that it is a partial evaluation of this property, when in preliminary laboratory tests [23,66], as it was suggested that in vivo studies would be the best way to assess the real potential bioactivity of materials [73,87].

To prove mineralization in biocompatibility studies in subcutaneous tissue, the von Kossa technique was used in some studies [60,61]. However, it has already been reported in the literature that this technique detects calcium and not calcium phosphate, with a complementary evaluation through other markers that indicate mineralization being recommended in some studies [88], such as alkaline phosphatase enzyme or even by X-ray diffraction, if available [89]. In a previous review [83], only bone implantation tests were included and subcutaneous implantation tests were excluded. In the present review, it was chosen to also include these studies because more recent investigations have evaluated the materials’ bioactive potential for their ability to stimulate mineralization in subcutaneous tissue using immunohistochemical techniques [65].

This study was carried out in an exploratory way to map the most commonly used methodologies to evaluate the bioactivity of bioceramic-based root canal sealers and obtain information on the materials’ properties according to the methodology used. The search strategy used included the term angiogenic according to a previous study [85], which was shown not to be a property to assess bioactivity directly, but to contribute to starting the repair process of the affected region [35]. In addition, as reported, a material would be considered bioactive when used in its clinical use environment, stimulating mineralization by the organism, having completed the material development phases with the knowledge of physicochemical properties, ex vivo tests, with culture of cells, animal tests, and finally clinical trials [13,90].

In the present review, an evaluation was carried out with a focus on methodologies for laboratory studies that could better predict the clinical performance of bioceramic-based root canal sealers in terms of bioactivity, which is important for subsequent randomized clinical studies that aim to clinically prove the performance of these materials.

## 5. Conclusions

According to the methodology used, the most prevalent methods to assess bioactivity in acellular form were the immersion of the material in Hank’s balanced salt solution, followed by surface observation with scanning electron microscopy and energy dispersive X-ray. In cell cultures, the chosen method was usually Alizarin Red staining, followed by the evaluation of alkaline phosphatase enzymatic activity and the use of molecular biology tests. In the animals’ subcutaneous tissue, the von Kossa histochemical technique was the most commonly used method to detect calcium deposition and, when evaluating bone tissue, the most commonly used histochemical technique was the use of hematoxylin–eosin.

## Figures and Tables

**Figure 1 life-12-01853-f001:**
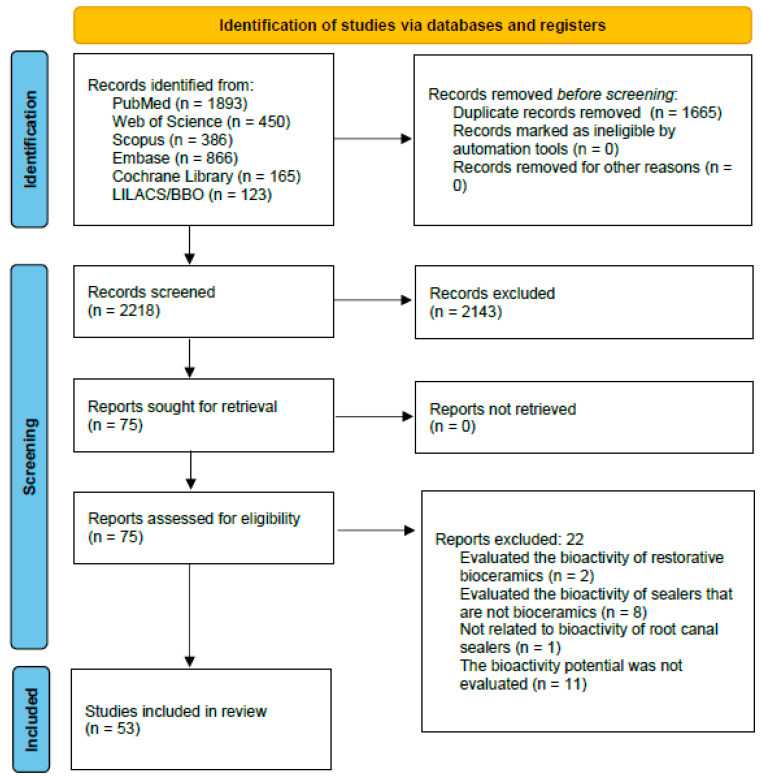
Search flowchart according to the PRISMA 2020 Statement.

**Table 1 life-12-01853-t001:** Studies that evaluated the potential for bioactivity in an acellular form in vitro.

Author and Year	Journal	Population	Tested Material	Objective	Bioactivity Analysis	Sample per Group	Duration	Main Results
[21] Benezra et al., 2018	Journal of Endodontics	Filled root canal	AH Plus (Dentsply DeTrey GmbH, Konstanz, Germany), MTA Fillapex (Angelus, Londrina, Brazil), BioRoot RCS (Septodont, Saint-Maur-des-Fossés, France), and Endoseal (Maruchi, Wonju-si, Gangwon-do, South Korea)	Evaluate the material-dentin interface of 3 bioceramic root canal sealers.	Immersion in Hank’s balanced saline solution and observation of the interface with confocal laser microscopy. Immersion in HBS ^a^ + FE/SEM ^b^ and EDS ^c^ (morphology and composition)	8	28 days	Root canal sealer penetration and interface characteristics were different for the materials tested. Confocal microscopy analysis showed a significant interfacial zone on the BioRoot RCS sealer. MTA Fillapex and BioRoot RCS exhibited the best cytocompatibility.
[14] Carvalho et al., 2017	Iranian Endodontic Journal	Dentin discs	AH-Plus (Dentsply) e Endo Sequence BC sealer (Endo Sequence, Brassler, Savannah, GA, USA)	Bioactivity through XRD ^e^ and demonstrate by SEM ^f^/EDS ^c^ with material immersed in SBF.	Immersion in SBF ^d^ + analysis by XRD ^e^ and demonstration by SEM ^f^ and EDS ^c^ (morphology and composition)	12	30 days	SEMf/EDSc analysis showed surface precipitates containing calcium and phosphorus. XRD analysis showed precipitates of Ca, Zr, Mg, Si, P, and Cl, suggesting bioactivity of BC sealer.
[22] Han et al., 2013	International Endodontic Journal	Dentin discs	White ProRoot MTA, (Dentsply), Bio dentine (Septodont, Saint, Maur des Fossés, France), Endo Sequence BC Sealer (Brasseler, Savannah)	Compare ProRoot MTA, Endo Sequence BC Sealer, and Bio dentine for their ability to produce apatite and cause Ca and Si incorporation into root dentin.	Immersion in PBS ^g^ + SEM-EPMA ^h^ (morphology and composition)	5	1, 7, 30, and 90 days	Formation of superficial precipitates of acicular morphology for Endo Sequence BC sealer with Ca/P 1.6 ratio.
[23] Jo et al., 2020	Nanomaterials	Sealer discs	AH Plus Jet (Dentsply), Well-Root ST (Vericom, Anyang, South Korea; WST), Endo Seal MTA (Maruchi, Won-ju, South Korea; EDS) and Nishika-BG (Nippon Shika Yakuhin, Shimonoseki, Japan; NBG)	Evaluate the acellular bioactivity of root canal sealers.	Immersion in HBSS or deionized water + FE-SEM ^b^ and analysis by EDX ^c^. (morphology and composition)	3	28 days	High amounts of calcium and phosphorus in the Well Root-ST groups indicated phosphate and calcium mineralization
[24] Oh et al., 2018	Materials	Filled root canal	New root canal sealer containing calcium zirconate and calcium silicate	Evaluate the sealing of materials through the dentin/sealer material interface.	Bacterial incubation + HBS ^a^ + FE-SEM ^b^ (morphology)	12	21 days	The canals filled with zirconate and calcium silicate cement had mineralization inside the dentinal tubules over 21 days. No significant differences were found between groups regarding endotoxin leakage.
[25] Sanz et al., 2021	Clinical Oral Investigations	Sealer discs	Bio-C Sealer ION+ (Angelus, Londrina, PR, Brazil), Endo Sequence BC Sealer HiFlow (Brasseler, Savannah), AH Plus (Dentsply)	Assess the mineralization potential of the root canal sealers.	HBSS ^a^ + SEM ^f^ and EDS ^c^ (morphology and composition)	3	48 h	Bio-C Sealer ION+ and BCHiF exhibited an irregular prismatic crystal structure on the surface.
[26] Siboni et al., 2017	International Endodontic Journal	Sealer discs	MTA Fillapex (Angelus), BioRoot RCS (Septodont, Saint-Maur-des Fosses, France), AH Plus (Dentsply), Pulp Canal Sealer (Kerr, Italy)	Evaluate the physicochemical properties of root canal sealers.	Immersion in HBSS ^a^ + SEM ^f^/EDX ^c^ and micro-Raman spectroscopy (morphology and composition)	10	28 days	BioRoot RCS and MTA Fillapex showed a layer of calcium phosphate
[27] Tanomaru-Filho et al., 2019	Brazilian Dental Journal	Sealer discs	MTA Fillapex^®^ (Angelus), Seal Apex (SybronEndo-Sybron Dental Specialties, Glendona, CA, USA), Sealer Plus (MKLife, Porto Alegre, RS, Brazil), AH Plus (Dentsply)	Evaluate the physicochemical properties and bioactive potential of Sealer Plus compared with MTA Fillapex, Seal apex, and AH Plus.	Immersion in PBS ^g^ + SEM ^f^ (morphology)		30 days	MTA Fillapex was the only material to demonstrate bioactive potential, with the formation of structures that suggest the presence of calcium phosphate
[28] Viapiana et al., 2014	Dental Materials	Sealer discs	S-Zr-micro (Araraquara Dental School, São Paulo State University, Brazil), ES-Zr-nano (Araraquara Dental School), ES-Nb-micro (Araraquara Dental School), ES-Nb-nano (Araraquara Dental School), AH Plus (Dentsply)	Characterize and evaluate the bioactivity potential of experimental materials.	Immersion in HBSS ^a^ + SEM ^f^ after setting and 28 days after EDS ^c^/XRD ^e^ and FT-IR ^i^ (morphology and composition)		1, 7, 14, 21, and 28 days	The experimental root canal sealers showed deposition of crystalline spherical structures of phosphate in calcium
[29] Wu et al., 2021	Materials Chemistry and Physics	Sealer discs	100TCS/0SPD, 90TCS/10SPD, 80TCS/20SPD, 60TCS/40SPD	Assess the apatite formation capacity.	SBF ^d^ + SEM ^f^/ EDS ^c^ (morphology and composition)	3	7 days	The root canal sealers tested induced the formation of apatite on the surface
[30] Cardoso et al., 2022	Journal of the Mechanical Behavior of Biomedical Materials	Sealer discs	Experimental bioactive sealer with bio glass. Experimental bioactive sealer with niobophosphate. MTA Fillapex (Angelus), AH Plus (Dentsply), Endo sequence BC Sealer (Brasseler), and Endofill (Dentsply)	Characterize and compare the bioactivity potential of the experimental bioactive sealers.	SEM-EDS, FTIR/ATR and XRD	2	28 days	The findings of this study showed that all sealers were somehow bioactive through the observation of hydroxyapatite precursors, with the exception of AH Plus and Endofill
[31] Huang et al., 2022	Journal of Dental Science	Filled root canal	Experimental bioactive sealer with bio glass, iRoot SP (Innovative Bioceramix Inc., Vancouver, BC, Canada)	Evaluate the biological properties of a novel bioceramic sealer.	Immersion in SBF for 28 days and analyzed through SEM	16	28 days	Dense hemispherical and plate-like hydroxyapatite crystals were observed on the surface of both the experimental sealer and iRoot SP

^a^ HBSS (Hank’s Balanced Salt Solution); ^b^ FE-SEM (Scanning Electronic Microscope—Field Emission); ^c^ EDX/EDS (Energy Dispersive X-ray Spectroscopy); ^d^ SBF (Simulated Body Fluid); ^e^ XRD (X-ray Diffraction); ^f^ SEM (Scanning Electron Microscope); ^g^ PBS (Phosphate Buffered Saline); ^h^ SEM-EPMA (Scanning Electron Microscope—Electron probe microanalysis); ^i^ FT-IR (Fourier Transform Infrared Spectroscopy).

**Table 2 life-12-01853-t002:** Studies evaluating the potential for cellular bioactivity in vitro.

Author and Year	Journal	Cell Type	Material Used	Dilution	Objective	Bioactivity Analysis	Sample (*n*)	Duration	Main Results
[21] Benezra et al., 2018	Journal of Endodontics	HGFs (human gingival fibroblasts cells)	AH Plus (Dentsply), MTA Fillapex (Angelus), BioRoot^™^ RCS (Septodont), and Endo Seal (Maruchi))	eluate 1:32	Evaluation of the activity of the ALP ^a^	ALP ^a^ activity	3	1 and 28 days	MTA Fillapex showed the highest value for alkaline phosphatase activity after 1 day of exposure and reduced after 28 days, and AH Plus and BioRoot RCS exhibited decreased bioactivity when compared with Endo Seal and MTA Fillapex.
[32] Bryan et al., 2010	Journal of Endodontics	MCT3-E1 Calvaria rat’s preosteoblast	Experimental calcium-silicate-based root canal sealer, AH Plus (Dentsply) Pulp Canal Sealer (SybronEndo, Orange, CA, USA); ProRoot White MTA (Dentsply)	1:10, 1:100, and 1:1000 aged for 6 weeks	Assess osteogenic potential	ALP ^a^ activity QuantiChrom ALP ^a^ assay kit (Bio-assay Systems, Hayward, CA, USA)	6	0, 4, 8, 12, 16 min	The activity of ALP on the AH Plus sealer was higher than the experimental silicate-based root canal sealer and higher than the pulp canal sealer. The formation of calcification nodules was observed for the experimental root canal sealer through observation of the presence of clusters of needle-shaped crystals
						Staining with von Kossa (mineralization of the extracellular matrix)		1 h silver nitrate and exposed 30 min in light	
						TEM ^b^ + stained for MCT-3E1 cells (experimental silicate-based root canal sealer group only)		28 days	
[33] Camps et al., 2015	Journal of Endodontics	hPDLCs (human periodontal ligament cells)	BioRoot RCS; (Septodont), Pulp Canal Sealer (SybronEndo)	Indirect contact with filled teeth immersed in culture medium	Evaluate the interactions with periodontal ligament bioactivity cells.	ELISA ^c^ (VEGF ^d^, FGF-2 ^e^ and BMP-2 ^f^)	3	2, 5, and 7 days	BioRoot RCS has fewer toxic effects on PDL cells and induced a higher secretion of angiogenic and osteogenic growth factors than Pulp Canal Sealer.
[34] Chang et al., 2014	Journal of Endodontics	hPDLCs (human periodontal ligament cells)	Apatite root canal sealer (Dentsply), iRoot SP (Innovative BioCreamix), MTA Fillapex (Angelus), Sealapex (SybronEndo)	Root canal sealer disc immersed in culture medium for 24 h	Assess the osteogenic potential and the signaling mechanism of biological activities	ALP ^a^ Activity Enzyme Assay ARS ^g^ + optical microscope Markers RT-PCR ^s^ (ON ^h^, OPN ^i^, OCN ^j^, RUNX2 ^k^, Osterix ^l^, β-actin ^m^)	4	7 and 14 days	Alkaline phosphatase activity on MTA Fillapex, Apatite Root Sankin, and iRoot SP increased at 7 and 14 days. Osteogenic potential at 7 days was higher on the tested bioceramic root canal sealers, and the formation of mineralized nodules was observed.
[35] Costa et al., 2016	Journal of Endodontics	hMSCs (human mesenchymal stem cells) Ex vivo Parietal bone of neonatal rats	ProRoot MTA (Dentsply), Bio dentine (Septodont), MTA Fillapex (Angelus), MTA Plus (Prevest Denpro Limited, Jammu City, India)	1:2, 1:20 dilution 21 days hMSCs and 7 days HUVECs	Assess the angiogenic and osteogenic responses.	ELISA ^c^-Activity of ALP ^a^	6	7, 14, and 21 days	Pro Root MTA and MTA Plus showed evident stimulatory effects on the proliferation of hMSCs, alkaline phosphatase activity, and ex vivo regeneration of bone defects when compared with the control groups
				1:20		Stained ALP ^a^ + SEM ^q^			
				1:5 and 1:20 dilutions		SEM ^q^			
[36] Dimitrova-Nakov et al., 2015	Dental Materials	A4 cells from the dental pulp (E 18 rats)	BioRoot RCS (Septodont), Pulp Canal Sealer (SybronEndo)	Root canal sealer discs	Evaluate the osteoinductive properties of BioRoot RCS compared with the pulp canal sealer	Immunocitycochemics BSP ^n^, COL-1 ^o^, and DMP-1 ^p^. Rabbit polyclonal primary antibodies and secondary antibodies were analyzed by broad-field indirect immunofluorescence	4	3, 7, and 10 days	BioRoot RCS promoted greater expression of BSP and DMP-1 on the cell surface. BioRoot RCS does not compromise mineralization potential in tested cells. BioRoot RCS was Von Kossa positive. However, the pulp canal sealer has not detected precipitated mineralization.
						Von Kossa to detect matrix mineralization			
[15] Garrido et al., 2021	BMC Oral Health	human apical papillary cells	UltraCal^®^ XS (Ultradent, South Jordan, UT, USA) ProRoot^®^ MTA (Dentsply) BioRoot RCS and Bio dentine (Septodont)	Root canal sealer discs	Assess the biological response of human apical papilla cells to the materials tested.	Cell adhesion assessed by SEM ^q^	5	24 h	Human apical papilla cells adhered to calcium silicate and calcium hydroxide-based materials, which is a good indicator of bioactivity.
[37] Giacomino et al., 2019	Journal of Endodontics	Murine osteoblast precursor cell line (IDG-SW3)	Endo Sequence BC Sealer (Brasseler), ProRoot ES (Dentsply), Roth (Roth International, Chicago, IL) and AH Plus (Dentsply)	Subtoxic concentrations	Evaluate the bioactivity of 2 bioceramic-based root canal sealers	ARS ^g^+ optical microscope	6	21 days	Endo Sequence BC Sealer and ProRoot ES were significantly more biocompatible and promoted osteoblastic differentiation. No signs of bioactivity were observed on the AH Plus and Roth sealers.
						Quantification of DMP-1p (Only DMP-1p) Cell Expression by Green Fluorescent Protein and Epifluorescence Microscopy RT-PCR ^s^ (DMP-1p, ALP ^a^, Phex ^r^)	12	7 days	
[38] Güven et al., 2013.	International Endodontic Journal	Human tooth germ stem cells (hTGSCs).	ProRoot MTA (Dentsply), iRoot SP (Innovative BioCreamix), Dycal (Dentsply)	Material in culture medium for 14 days	To compare the effect of MTA and iRoot SP on hard tissue deposition and odontogenic differentiation in human tooth germ stem cells.	Activity of ALP ^a^	6	14 days	MTA and iRoot SP induced hTGSC differentiation into odontoblast-like cells, but MTA might provide more inductive potential and hard tissue deposition compared with iRoot SP.
						enzyme assay			
						Immunocytochemistry (COL1 ^o^ and DSP ^a2^) with antibodies and fluorescence microscope			
						RT-qPCR ^t^ (COL1 ^o^ e DSPP ^a2^)			
						Von Kossa + optical microscope			
[39] Hakki et al., 2013	International Endodontic Journal	Immortalized murine cementoblast cell line (OCCM-30)	AH Plus (Dentsply), Hybrid Root Seal (Sun Medical Co. Shiga, Japan), Real Seal (SybronEndo), SimpliSeal (DiscusDental, LLC, Culver City, CA, USA), TECH Bio sealer Endo (Isasan, Italy)	1:1, 1:2, 1:4	To investigate the gene expression of proteins associated with mineralized tissue formation in cementoblasts	RT-qPCR ^t^ (BSP ^n^, OCN ^j^, Runx2 ^k^, COL1 ^o^, ALP ^a^)	6	24 h	Tech Biossealer Endo decreased mRNA expression for COL1, ALP, BSP, and OCN. SimpliSeal and AH Plus resulted in a more favorable response to cementoblasts because of their regulation potential on the mineralized tissue-associated protein’s mRNA expressions.
[40] Jing et al., 2019	Journal of International Medical Research	hPDLSCs (human periodontal ligament cells)	BioRoot RCS (Septodont), AH-Plus (Dentsply), C-Root (experimental material)	Apical third of roots in cell culture medium	Evaluate the osteogenic potential of an experimental silicate-based root canal sealer	ARS ^g^ test + phase contrast microscope.	6	14 days	The experimental root canal-filling material C-Root has similar in vitro cytocompatibility to BioRoot RCS and better osteogenic potential than AH Plus.
						ALP ^a^ (staining assay)		14 days	
						RT-qPCR ^t^ (ALP ^a^, OCN ^j^, RUNX2 ^k^, DMP-1p)		7 and 14 days	
[23] Jo et al., 2020	Nanomaterials	hPDLCs (cells of the human periodontal ligament) HUVECs (human umbilical cord endothelial cells)	AH Plus Jet (Dentsply), Well-Root ST (Vericom, Anyang, South Korea; WST), Endoseal MTA (Maruchi) and Nishika-BG (Nippon Shika Yakuhin, Shimonoseki, Japan)	1:2 diluted Dilution (50%) Dilution (25%)	cell bioactivity Angiogenic gene expression. Angiogenic Tube Formation Assay	ALP ^a^ (Staining Assay)	3	3, 7, and 21 days	All bioactive root canal sealers released calcium ions, while Nishika Canal Sealer BG released 10 times more silicon ions than the other bioactive root canal sealers. Under the cytocompatible extraction range, Nishika BG showed prominent cytocompatibility, osteogenecity, and angiogenecity compared with other sealers in vitro.
						ARS ^g^ (Alizarin Red staining)		21 days	
						RT-PCR ^s^ (DMP-1p, RUNX2 ^k^ and OSX ^l^)		3, 7, and 21 days	
						RT-PCR ^s^ (VEGF ^d^, PDGF-BB ^u^, bFGF ^v^)		3, 7, and 21 days	
						Number of nodules and circles kit		3, 7, and 21 days	
[41] Lee et al., 2019	Journal of Endodontics	MC3T3-E1 cells.	AH Plus (Dentsply), MTA Fillapex (Angelus), and Endo Sequence BC Sealer (Brasseler)	1/10 culture medium extract and LPS (100ng/mL)	Evaluation of the osteogenic potential of AH Plus, MTA Fillapex and Endo Sequence BC Sealer	RT-PCR ^s^ and real-time PCR (+Escherichia coli LPS) for osteogenic markers ALP ^a^ and OCN ^j^	6	0, 1, and 2 days.	MTA Fillapex and Endo Sequence BC showed strong cell viability compared with AH Plus. AH Plus, MTA Fillapex, and Endo Sequence BC decreased the levels of LPS-induced inflammatory mediators. The expression of osteogenic marker genes, alkaline phosphatase activity, and mineralized nodule formation decreased with LPS treatment. However, AH Plus and bioceramic-based sealers increased the osteogenic potential reduced by LPS treatment
						ALP ^a^ Staining + photographed + spectrophotometer	6	7 days	
						ARS ^g^	6	14 days	
[42] López-García et al., 2019	Materials	hPDLSCs (Human Periodontal Ligament Cells)	Bio-C Sealer (Angelus), Total Fill BC Sealer (FKG Dentaire SA, La-Chaux-de-fonds, Switzerland) e AH Plus (Dentsply)	1:1 dilution	Assess mineralization potential	ARS ^g^ (*Alizarin Red Assay*)		21 days	Bio-C Sealer and Total Fill BC Sealer demonstrated better cytocompatibility in terms of cell viability, migration, cell morphology, cell attachment, and mineralization capacity than AH Plus.
[43] Lopez-Garcia et al., 2020	Clinical Oral Investigations	hPDLSCs (human periodontal ligament cells)	Endo Sequence BC Sealer (Brasseler USA, Savannah, GA, USA), Endo Seal MTA (Maruchi), CeraSeal (Meta Biomed Co., Cheongju, Korea)		Evaluate the biological properties of 3 calcium-silicate-based root canal sealers	RT-qPCR ^t^ for (CEMP-1 ^w^, CAP ^x^, and ALP ^a^)	3	3, 7, 14, and 21 days	ALP-3 and 7 days Endo Sequence BC Sealer and Ceraseal displayed higher cell viability, cell attachment, cell migration rates, and ion release rates than Endoseal. Ceraseal and Endo Sequence BC Sealer exhibited significantly more gene expression and mineralization capacity than Endoseal.
						ARS ^g^ + spectrophotometer	6	21 days	
[44] Mestieri *et al.*, 2015	Journal of Applied Oral Science	hDPCs (human dental pulp cells)	MTA Fillapex (Angelus, Londrina, PR, Brazil) and MTA Plus (Avalon Biomed Inc., Sarasota, FL, USA) and FillCanal (Technew, Rio de Janeiro, RJ, Brasil)	1:2, 1:3, and 1:4	Assess biocompatibility and bioactivity	Atividade da ALP ^a^ kit (Labtest Diagnóstica, Lagoa Santa, MG, Brazil).	3	1 and 3 days	>In the MTA Plus group, the cells’ ALP activity was similar to positive control in one and three days of exposure to the material. MTA Fillapex and Fill Canal sealer groups demonstrated a decrease in ALP activity when compared with positive control at both periods of cell exposure.
[45] Oh et al., 2020	Materials	hPLCs (human periodontal ligament stem cells)	CeraSeal (MetaBiomed, Cheongju, Korea), Endo Seal TCS (Maruchi), and AH-Plus (Dentsply)	Eluate-disc of material in culture medium	Assess osteogenic potential, gene expression.	RT-qPCR ^t^, (ALP ^a^, RUNX2 ^k^, and OCN ^j^)	4	1, 3, 7, and 14 days	Endo Seal TCS showed better osteogenic potential than the other tested materials.
						ARS ^g^ + ALS + optical microscope		14 days	
						SEM ^q^ (cell adhesion)			
[46] Rodríguez-Lozano et al., 2019	Dental Materials	Human periodontal ligament stem cells (hPLSCs)	GuttaFlow Bio seal (Coltène, Altstatten, Switzerland), GuttaFlow2 (Coltène), MTA Fillapex (Angelus), AH Plus (Dentsply)	Undiluted, 1:2, and 1:4	Potential cementogenic in contact with human periodontal ligament stem cells (hPDLSCs)	Through RT-qPCR ^t^ and CEMP1 ^w^, CAP ^x^, BSP ^y^, AMBN ^z^, AMELX ^a1^, ALP ^a^)	5	7 and 21 days	When hPDLSCs were cultured with GuttaFlow Bio seal-conditioned media, qPCR assays and IF showed a higher level of AMELX, AMBN, CEMP1, and CAP expression than the control, whereas no such expression was observed in the other sealers.
						Immunofluorescence analysis of protein expression (CP1 ^w^ and CAP ^x^) and observation by confocal laser microscopy			
[47] Rodríguez-Lozano et al., 2020	International Endodontic Journal	Periodontal ligament stem cells (hPLSCs)	Endo Sequence BC Sealer HiFlow (Brasseler), Endo Sequence BC Sealer (Brasseler) and AH Plus (Dentsply)	Undiluted, 1:2, and 1:4	To evaluate the biological effects of Endo Sequence BC HiFlow compared with Endo Sequence BC Sealer and an AH Plus epoxy resin-based root canal sealer.	ARS ^g^	3	7 and 21 days	Endo Sequence BC HiFlow, Endo Sequence BC Sealer, and Osteodiff produced significantly more calcium deposits than the control group alone after 21 days of culture. The greatest mineralization capacity was seen with the Endo sequence BC group compared with Endo sequence BC Sealer HiFlow and Osteodiff.
						RT-qPCR ^t^ (ALP ^a^, CEMP-1 ^w^, and CAP ^x^)		7 days	
[48] Salles et al., 2012	Journal of Endodontics	Human osteoblast cells (Saos-2 line ATCC HTB-85)	MTA Fillapex (Angelus), Epiphany SE (SybronEndo), Zinc oxide–eugenol root canal sealer		Assess bioactivity	Activity of ALP ^a^	6	21 days	The ALP activity increase was significant in the MTA Fillapex group. MTA Fillapex presented the highest percentage of ARS-stained nodules. SEM/EDS analysis showed hydroxyapatite crystals only in the MTA Fillapex and control groups. However, crystallite morphology and chemical composition were different from the control group.
						ARS ^g^			
[25] Sanz et al., 2021	Clinical Oral Investigations	Human periodontal ligament cells (hPDLCs)	Bio-C Sealer ION+ (Angelus), Endo Sequence BC Sealer HiFlow (Brasseler), AH Plus (Dentsply)	1:2 e 1:4	Assess mineralization potential	HBSS (Hank’s balanced salt solution) + SEM ^q^ and EDS (energy-dispersive spectroscopy),	3	21 days	The inconsequence BC HiFlow group showed an upregulation of CAP (*p* < 0.01), CEMP1, ALP, and RUNX2 (*p* < 0.001) compared with the negative control, while the Bio-C Sealer ION+ group showed an upregulation of CEMP1 (*p* < 0.01), CAP, and RUNX2 (*p* < 0.001). Both groups also exhibited a greater mineralization potential than the negative and positive controls
						ARS ^g^		21 days	
						RT- qPCR ^t^ (CEMP1 ^w^, CAP ^x^, ALP ^a^, RUNX2 ^k^)		21 days	
[49] Seo et al., 2019	Materials	Human stem cells 3 hDPSCs	AH Plus (Dentsply), Endo Sequence BC Sealer (Brasseler), BioRoot RCS (Septodont) Endoseal MTA (Maruchi)		Evaluate cytotoxic effects and mineralization activity	ARS ^g^	4	15 days	Endo Sequence BC Sealer, BioRoot RCS and Endo Seal MTA exhibited increased mineralization activity compared with AH Plus
[50] Tanomaru-Filho et al., 2017	International Endodontic Journal	Saos-2 human osteoblast-like cells (ATCC HTB-85)	experimental TSC/Ta2O5, Neo MTA Plus (Avalon Biomed Inc.), MTA (Angelus)	1: 8 dilution	Assess biocompatibility and formation of mineralized nodules	ALP ^a^ (kit comercial)	(18 per group)	1, 3, and 7 days	All materials induced the production of mineralized nodules, being higher with MTA Plus.
						ARS ^g^	(12 per group)	21 days	
[51] Washington et al., 2011	Journal of Endodontics	Rats primary osteoblasts	Generex A e Generex B (calcium silicate based), Capasio (calcium-phospho-alumino silicate based), Ceramicrete-D (magnesium phosphate based)		Assess osteogenic potential	Nodules mineralized by SEM ^q^	5	7 and 14 days	Generex A was the only material that supported the growth of primary osteoblasts.
[52] Wu et al., 2020	Stem Cells International	BMSCs—bone marrow mesenchymal stem cells (rats)	iRoot^®^ SP (Innovative BioCeramix Inc.)	20mg/mL Culture medium	Effect of iRoot SP on BMSCs and the molecular mechanisms of any identified effects.	ALP ^a^- ELISA ^c^	3	0, 3, and 7 of exposure to the material	iRoot SP-conditioned medium significantly elevated osteo/odontogenic differentiation of BMSCs via the MAPK and NF-κB cascades
						ARS ^g^ + microscope			
						quantification ARS ^g^			
						Western Blot (DSPP ^a2^, OPN ^i^, Runx2 ^k^, OSX ^l^)			
						Real-Time PCR and RT-qPCR ^t^ (OSX, ALP ^a^, RUNX2 ^k^, OPN ^j^, DSPP ^a2^)			
						Immunofluorescence staining (ALP ^a^, RUNX2 ^k^)			
[53] Zhang et al. 2010	Journal of Endodontics	Human osteoblasts (MG63 cells)	iRoot^®^ SP (Innovative Bio-Creamix Inc.)	1:1, 1:2, and 1:4	Evaluate the effects of gene expression related to mineralization during hard tissue formation	ARS ^g^	6	1, 3, and 6 days	iRoot SP up-regulated COL I, OCN, and BSP messenger RNA expression after 3 and 6 days. In the presence of iRoot SP, MG63 cells can produce more mineralized matrix gene and protein expression.
						Elisa ^c^ (COL1 ^o^, BSP ^n^)			
						RT-PCR ^s^ (BSP ^n^, COL1 ^o^, OCN ^j^, OPN ^i^)			
[54] Zordan-Bronzel et al. 2019	International Endodontic Journal	Saos-2 human osteoblast-like cells (ATCC HTB-85)	Experimental (Araraquara Dental School, Brazil), Total Fill BC (FKG), AH Plus (Dentsply)	dilutions 1: 2, 1: 4, 1: 8, 1: 16, and 1: 32	Assess the potential to induce mineralization	ALP ^a^ Kit test and ARS ^g^	3	7 days	Significantly greater mineralized nodule production was observed for Total Fill BC and the experimental sealer when compared with the control group.
[55] Mann et al., 2022	Journal of Endodontics	Human periodontal ligament (HPDL) fibroblasts	Endo sequence BC Sealer HiFlow (Brasseler), Endo sequence BC Sealer (Brasseler), and AH Plus (Dentsply)	Undiluted	Evaluate biological properties related to mineralization genes	Alkaline phosphatase (ALP), runt-related transcription factor 2 (RUNX2), and osteocalcin (OC)	3	24 h	No significant differences in the mRNA expression of the studied genes were found among the tested sealers.

^a^ ALP (Alkaline Phosphatase Enzyme); ^b^ TEM (Transmission Electronic Microscope); ^c^ ELISA (Enzyme Immunosorbent Assay); ^d^ VEGF (Vascular Endothelial Growth Factor); ^e^ FGF-2 (Fibroblast Growth Factor 2); ^f^ BMP-2 (Bone Morphogenetic Protein-2); ^g^ ARS (Alizarin Red Stain); ^h^ ON (Osteonectin); ^i^ OPN (Osteopontin); ^j^ OCN (Osteocalcin); ^k^ RUNX-2 (transcription factor 2 related to runt); ^l^ OSX (Marker for osteogenic genes); ^m^ Beta-Actin (Actin); ^n^ BSP (Bone Sialoprotein); ^o^ COL -1 (Collagen type 1 protein); ^p^ DMP-1 (Acid Phosphoprotein 1); ^q^ SEM (Scanning Electron Microscope); ^r^ Phex (Neutral Phosphate Regulating Endopeptidase); ^s^ RT-PCR (Reverse Transcription Polymerase Chain Reaction); ^t^ RT-qPCR (Quantitative Reverse Transcriptase Polymerase Chain Reaction); ^u^ PDGF-BB (Growth Factor); ^v^ bFGF (Fibroblast Growth Factor); ^w^ CEMP-1 (Recombinant Cement Protein); ^x^ CAP (cementum binding protein); ^y^ BSP (Bone Sialoprotein); ^z^ AMBN (Ameloblastin); ^a1^ AMELX (Amelogenin); ^a2^ DSPP (Dentin Sialophosphoprotein).

**Table 3 life-12-01853-t003:** In vivo studies and their main results.

Author and Year	Journal	Population	Material	Objective	Bioactivity Analysis	Number per Period	Periods	Main Results
[59] Almeida, et al., 2019.	International Endodontic Journal	Wistar rats	MTA Fillapex (Angelus), MTA Fillapex C3A, MTA Fillapex C3A + Ag 1%, MTA Fillapex C3A + Ag 5%, Endo Sequence BC Sealer (Brasseler)	To evaluate the bone tissue reaction in rats to an MTA-based dental root sealer and the effect of adding calcium aluminate and silver particles at various concentrations.	Histological-bone tissue formation. Stained with hematoxylin–eosin	5	7, 30, and 90 days	MTA Fill apex showed the partial formation of mineralized tissue barrier, but Endo Sequence BC Sealer had full formation.
[68] Assmann et al., 2015.	Journal of Endodontics	Wistar rats	MTA Fillapex^®^ (Angelus)	Evaluate bone tissue reaction to MTA Fillapex compared with an epoxy resin-based material in rat femurs.	Histological bone tissue formation.	5	7, 30, and 90 days	Bone formation was similar to the control group and bone barrier formation was found in 90 days.
[60] Delfino, et al., 2020.	Scientific reports	Wistar rats	GuttaFlow^®^ bio seal (Coltene), MTA Fillapex^®^ (Angelus)	Evaluation of the immunoinflammatory response and bioactive potential of materials tested in subcutaneous tissue and mineralization VEGF (endothelial growth factor)	Histochemical evaluation by the hematoxylin–eosin method + optical microscope and von Kossa + analysis with polarization microscope Immunohistochemistry	5	7, 15, 30, and 60 days	The capsules of the materials evaluated were von Kossa positive and thinner than those of the Endo fill group. GuttaFlow Bio seal had lower values than MTA Fillapex and lower than Endo fill.
						5	7, 15, 30, 60, and 90 days	
[61] Gomes-Filho et al., 2009.	Journal of Endodontics	Wistar rats	Endo-CPM-Sealer (EGEO S.R.L.; Buenos Aires, Argentina), MTA (Angelus)	Assess tissue response and mineralization.	Histochemical reaction by the Von Kossa method and analysis with a Hematoxylin–eosin polarization microscope.	6	7, 15, 30, 60, and 90 days	Mineralization and birefringent granulations under polarized light were detected in all materials.
[69] Gomes-Filho et al., 2012	Dental Traumatology	Wistar rats	MTA Fillapex^®^ (Angelus)	Assess the reaction of the subcutaneous tissue and the ability to stimulate mineralization.	Histochemical reaction by the Von Kossa method and analysis with a Hematoxylin–eosin polarization microscope.	6	7, 15, 30, 60, and 90 days	Mineralization and birefringent granulations under polarized light were observed with all materials. MTA Fillapex^®^ was biocompatible and stimulated mineralization.
[62] Gomes-Filho et al., 2016	Brazilian Oral Research	Wistar rats	Sealapex^®^ (SybronEndo) MTA Fillapex (Angelus)	Evaluate the influence of Diabetes Mellitus on tissue response and mineralization using MTA Fillapex^®^	von Kossa, Calcein fluorescent marker, ARS, and tetracycline hydrochloride	6	7 and 30 days	MTA Fillapex was von Kossa positive in all groups at 7 and 30 days. Diabetes Mellitus did not influence tissue response or mineralization.
[63] Hoshino et al., 2021	Restorative Dentistry and Endodontics	Holtzman rats	NeoMTA Plus (Avalon Biomed Inc.) MTA Fillapex^®^ (Angelus)	Evaluate tissue response induced by NeoMTA Plus compared to MTA Fillapex and bioactivity.	Histological observation using hematoxylin–eosin, von Kossa’s histochemical method, contrasted with Syrian red (picrosirius red), and observation with a polarized light microscope.	5	7, 15, 30, and 60 days	The materials showed biocompatibility and bioactive potential. Positive von Kossa structures.
[70] Okamura et al., 2020	Materials	Dog teeth	Bio-C Sealer (Angelus)	Investigate the histological response in endodontically treated dog teeth	Hematoxylin–eosin staining protocol observed through polarized microscope and radiographic analysis.	18	28 and 90 days	At 28 and 90 days, the presence of immature periodontal ligament fibers and a thick cementum layer at the apex was observed with the two materials, demonstrating bioactive potential.
[58] Petrović et al., 2021	Acta Veterinaria	Rabbits’ teeth	Experimental calcium silicate and Experimental calcium silicate + hydroxyapatite	Evaluate the periradicular inflammatory reaction and calcified tissue formation after root canal sealer implantation.	Hematoxylin–eosin staining protocol observed through a polarized microscope	6	28 days	Hydroxyapatite–calcium Silicate >MTA and calcium silicate with respect to the continuity of neo formed calcified tissue. The experimental root canal sealers showed minimal tissue inflammatory response, similar to MTA.
[64] Santos et al., 2021.	Biomedicines	Wistar rats	TotalFill BC Sealer (FKG), TotalFill BC Sealer HiFlow (FKG)	Evaluate the biocompatibility and bioactivity potential of two hydraulic calcium-silicate-based root canal sealers in subcutaneous tissue	Histochemical reaction by the Von Kossa method and analysis with hematoxylin–eosin staining protocol observed through a polarized microscope		8 and 30 days	Mineralization potential was only observed with TotalFill BC Sealer and TotalFill BC Sealer HiFlow groups. When compared with the control, TotalFill BC Sealer and TotalFill BC Sealer HiFlow were considered biocompatible and showed potential for bioactivity when deployed in subcutaneous tissues.
[65] Silva et al., 2020.	Journal of Endodontics	Holtzman rats	Bio C-Sealer (Angelus), Sealer Plus BC (MK Life, Porto Alegre, Brazil)	Evaluate the biocompatibility and bioactive potential of 2 bioceramic-based dental root sealers with sealers based on epoxy resin. (Subcutaneous)	Von Kossa histochemical reaction and analysis with polarizing microscope Osteocalcin immunohistochemical detection Hematoxylin–eosin staining protocol observed through polarized microscope	6	7, 15, 30, and 60 days	Sealer Plus BC and Bio C-sealer showed Von Kossa positive structures and osteocalcin immunopositive cells, demonstrating bioactivity potential.
[66] Silva et al., 2021.	International Endodontic Journal	Holtzman rats	Two experimental dental root sealers (CE-1 and CE-2)	Biocompatibility and bioactive potential of two experimental dental root sealers (CE-1 and CE-2) and osteocalcin detection. (Subcutaneous)	von kossa histochemical reaction and analysis with hematoxylin–eosin polarization microscope and optical microscope immunohistochemical detection of osteocalcin	6	7, 15, 30, and 60 days	Experimental dental root sealers had bioactive potential observed by the detection of osteocalcin. TotalFill BC Sealer, CE-1, and CE-2 showed positive Von Kossa structures.
[67] Viola Viana et al., 2012.	Journal of Biomedical Materials Research	Holtzman rats	Experimental dental root sealers based on MTA (Angelus), Portland cement (Votorantin, MG, Brazil)	Subcutaneous tissue reaction by morphology, histochemistry, immunohistochemistry, and quantitative analysis of inflammatory cells.	Histochemical reaction by hematoxylin–eosin + Von Kossa optical microscope and analysis with polarization microscope. Immunohistochemistry for osteopontin	5	7, 14, 30, and 60 days	Experimental MTA-based sealers exhibited similar biological response to MTA and Portland root canal sealers and showed osteopontin detection and positive Von Kossa.
[71] Zhang et al., 2015	Dental Materials Journal	Wistar rats	AH Plus (Dentsply), ProRoot MTA (Dentsply), iRoot SP (Innovative BioCreamix)	Evaluate the subcutaneous and bone reaction to iRoot SP in vivo.	Histological evaluation Hematoxylin–eosin and optical microscope	12	7, 30, and 60 days	iRoot SP and MTA showed a similar inflammatory reaction and were considered biocompatible with subcutaneous and intraosseous tissues in rats.
[72] Zmener et al., 2020	Revista de la Asociación Odontológica Argentina	Wistar rats	Bio-C Sealer (Angelus), MTA Densell (Densell, Buenos Aires, Argentina)	Compare the biocompatibility of Bio-C Sealer and MTA Densell implanted in bone tissue of Wistar rats	Hematoxylin–eosin and optical microscope	10	7, 30 and 90 days	Bio-C Sealer and MTA Densell behaved as biocompatible and osteoinductive.
[73] Belal et al., 2022	Clinical Oral Investigations	Wistar rats	Endo sequence BC Sealer (Brasseler), MTA Fillapex (Angelus), Nishika Canal Sealer BC (Nippon Shika Yakuhin, Japan)	Compare the three bioceramic-containing root canal sealers in terms of their in vivo apatite-forming ability	Ultrastructure and elemental composition using an electron probe microanalyzer (EPMA1601; Shimadzu) equipped with observation functions for scanning electron microscopy and wavelength-dispersive X-ray spectroscopy	4 for each sealer	28 days	The in-vivo-implanted Endo sequence BC displayed apatite-like spherulites on the surface, whereas the other bioceramic sealers displayed no spherulites.

## Data Availability

Acquired data are available on the *Open Science Framework* through the link https://osf.io/3jdqu (4 May 2021).

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
