# Peer review of "Bioactivity Potential of Bioceramic-Based Root Canal Sealers: A Scoping Review"

_life, 2022, doi:10.3390/life12111853_

Round 1

Reviewer 1 Report

With these search criteria, perhaps the PICO question should be rephrased. The title is not in line with it either. 

Method has to be reviewed. 

Some references are duplicated. 

Author Response

Thank you for your suggestion! Since our paper is a scoping review, we followed the Joanna Brigs Institute recommendations (Munn et al. 2018) which preconize the PCC question (population-context-concept) instead of the PICO question. That is the reason why the research question was formulated in that way. This issue was clarified within the revised manuscript

Finally, all references were checked, and the duplicated references were removed.

Reviewer 2 Report

Dear Authors, 

you made a great work. However, some improvements are mandatory before acceptance. 

Author Response

After suggestions we performed some changes in the manuscript, We hope that the changes and explanations can improve readability. If more revisions prove necessary, we are entirely at your disposal.

Reviewer 3 Report

Reviewer’s comments

Ms.ID life-1968786

Title:  Bioactivity potential of bioceramic-based root canal sealers: a scoping review

Authors: Mauro Schmitz Estivalet, Lucas Peixoto de Araújo, Felipe Immich, Adriana Fernandes da Silva, Nadia de Souza Ferreira, Wellington Luiz de Oliveira da Rosa, Evando Piva

General Comments:

  The aim of this study is to synthesize and map key concepts related to the bioactivity analysis of bioceramic-based root canal sealers.  

  Before the acceptance to life, I want the authors to tell me to the followings;

1. Do the authors think that how methodologies are best to study “bioactivity”? 

Author Response

Answer: There is still no clear consensus on which method is more reliable. However, bioactivity should be further studied in ‘in vivo’ conditions through von Kossa staining techniques associated with mineralization markers such as alkaline phosphatase enzyme and x-ray diffraction. Moreover, future studies should investigate the impacts of bioceramic sealers on the human periodontal ligament cells and associate the findings with future results from randomized clinical trials that investigate clinical outcomes using this type of sealer. Thank you for your contribution!

REFERENCES:

Munn Z, Peters MDJ, Stern C, Tufanaru C, Mcarthur A, Aromataris E. 2018. Systematic review or scoping review ? Guidance for authors when choosing between a systematic or scoping review approach. :1–7.